# Open Knowledge about Slaughter on the Internet—A Case Study on Controversies

**DOI:** 10.3390/ani7120101

**Published:** 2017-12-18

**Authors:** Anne Algers, Charlotte Berg

**Affiliations:** 1Faculty of Education, Department of Education, Communication and Learning, University of Gothenburg, Box 300, SE40530 Gothenburg, Sweden; 2Faculty of Veterinary Medicine and Animal Science, Department of Animal Environment and Health, the Swedish University of Agricultural Sciences, Box 234, SE532323 Skara, Sweden; lotta.berg@slu.se

**Keywords:** animal welfare resilience, focus group, inclusiveness, learning, open educational resource, slaughter

## Abstract

**Simple Summary:**

Animal products are consumed by a large majority of the global population, yet public knowledge about animal handling and welfare during the slaughter process is limited. An open educational resource about slaughter, called “Animal welfare at slaughter and killing” has been openly available on the Internet since 2012. The resource includes learning objectives, 650 webpages, 800 illustrations, 150 video clips, self-tests with feedback and a series of take-home messages. The resource is designed to not only be relevant to the primary target group, i.e., the abattoir staff, but also to anyone with an interest in the topic. A study was conducted to evaluate the use and impact of this educational resource with participants from slaughterhouses, universities, authorities and non-governmental organizations (NGOs). Focus group sessions were video recorded and analysed using an interpretive thematic analysis. Improved knowledge among consumers may lead to more well-founded decisions at purchase of meat and improved awareness among citizens to increase public pressure to improve animal handling at slaughter.

**Abstract:**

Knowledge about slaughter of animals for human food is often perceived as controversial and therefore not made widely available. An open educational resource on the Internet about the slaughter of animals has created tension at launch but also resolved tension. Aiming to explore how this resource at the boundary between academia and society is perceived, a study was carried out with participants from slaughterhouses, universities, authorities and NGOs. Focus group sessions were video recorded and transcripts were coded using an interpretive thematic analysis. The results show that an open educational resource in addition to contributing to learning and awareness raising can also induce dialogue (and thus resolve tension) about animal welfare and contribute to animal welfare resilience. Our results also indicate that participants had diverse opinions about the influence of multimedia on attitudes towards animal slaughter. The use of additional instruments such as comment fields may lead to more knowledgeable citizens and socially robust knowledge, but has to be carefully weighed against the risk of false or fake data.

## 1. Introduction

Boundaries are social constructions that define who is included and excluded from interactions [1]. Boundary objects can connect actors from different social worlds with different agendas to create common meanings, since such objects are “both plastic enough to adapt to local needs and the constraints of the several parties employing them, yet robust enough to maintain a common identity across sites” [2]. Star and Griesemer [2] emphasize the introduction of an object to achieve boundary activities whereas Engeström et al. [3] are more focused on the process when generating boundary objects through boundary activities. The latter view is more in compliance with open learning theories, where actors create the boundary object that invites users to reuse and further develop the object itself. In this article, we study the creation and use of a digital learning resource (i.e., an object) at the boundary between academia and society.

In a recent Special Eurobarometer [4], citizens expressed concern and interest in animal welfare and there was a more pronounced desire to know about animal welfare than seen during a previous survey 10 years ago [5]. During the last two decades, EU legislation on animal welfare has changed as a result of the adoption of the Amsterdam Treaty [6], which states that animals can feel pain and suffer, and the Lisbon Treaty [7], which states that, since animals can suffer, we need to pay full regard to their welfare.

This resulted in the European Animal Welfare Strategy that “puts animal welfare on equal footing with other key principles mentioned in the same title i.e., to promote gender equality, guarantee social protection, protect human health, combat discrimination, promote sustainable development, ensure consumer protection, protect personal data” [8]. An accompanying action plan [9] described the challenges of raising awareness about animal welfare among all members of society. 

Animal welfare at slaughter, including the occurrence of slaughter at all, is a particularly controversial area covered by extensive legislation at the EU level, mainly through EC Regulation 1099/2009, which includes requirements in relation to training courses and certificates of competence for slaughterhouse staff. However, how to satisfy the citizens’ interest of more knowledge about animal welfare at slaughter is not well researched.

In Sweden, learning material about animal welfare at slaughter and killing was created by a team of researchers from the Swedish University of Agricultural Sciences (SLU) in collaboration with individual experts, slaughterhouses and non-governmental organizations (NGOs). The learning material was named DISA—an acronym for “Djurvälfärd i samband med Slakt och annan Avlivning” in English: “Animal welfare at slaughter and killing”—and included learning objectives, 650 webpages, 800 illustrations, 150 video clips, formative assessment with feedback and take-home messages. During the creation phase, the team had on-going discussions with external agents, who reviewed DISA and improved the practical handling details. The learning objectives included in the material are clearly expressed at the beginning of each chapter, to make it easier for the user to identify the central ‘take home messages’ and the level of understanding expected. A summative assessment is organised by a national organization and a certificate of competence is issued for slaughterhouse staff who have passed the examination.

Publishing photos and video footage showing slaughter is controversial because such pictures can be perceived as upsetting or aversive. The slaughter industry’s umbrella organizations were concerned that images from the resource would be used by animal rights groups to discredit abattoirs and the industry in general. This concern was, however, not voiced by the individual slaughterhouses.

Hence, these slaughter industry organizations suggested requiring a password to access the material. After being put under pressure by the umbrella organizations, the Swedish Board of Agriculture (the main funder of the DISA development process) succumbed to the pressure and suggested a system based on restricted access to the learning resource, which in practice meant disregarding a signed contract with SLU about open access to the educational resource. 

The aim with DISA was twofold: to support local efforts to increase understanding of relevant animal welfare regulations and to provide free access for anybody interested in gaining knowledge about animal welfare at slaughter and killing. The different opinions among stakeholders created a conflict, as a result of pressure from the umbrella organizations and the Swedish Board of Agriculture on SLU to stop the launch of the openly accessible learning material. However, the researcher responsible for DISA at SLU informed the lawyer at the university about the conflict and the lawyer wrote to the involved stakeholders: “In the choice between meeting the requirements from upset stakeholders and safeguarding academic integrity, a university must always choose the latter”. Consequently, DISA was launched and is openly available at http://disa.slu.se/ [10]. An English translation is available, and a Chinese translation is being discussed. 

In November 2013, the European Commission also published a document outlining “Opening up Education”. It included an action plan and encouraged member states and higher education institutions to stimulate open access policies for higher education publicly funded educational materials. It also encouraged the inclusion of digital content and open educational resources (OER) whose copyrights would belong to public authorities [11]. 

The learning process benefits when every member of society is included, especially when dealing with complex issues such as animal welfare. The Special Eurobarometer [4] demonstrated that the younger generation (including people still studying) is more likely to be interested in receiving more information about animal welfare. Furthermore, it is crucial that people have access to research in order to adapt to emerging challenges. For example, the EU has recently decided to establish a platform on animal welfare for dialogue between scientists, industry stakeholders, policy makers and society [12]. 

The Eurobarometer concludes that a majority of the 27.672 respondents find that the EU should do more to promote a greater awareness of animal welfare internationally and to establish standards that are recognised across the world [1]. 

The aim of this study was to investigate if an OER (like DISA) about animal welfare at slaughter can be regarded as a boundary object that can handle: (a) inclusiveness, (b) learning and awareness raising, (c) controversies, (d) accuracy and relevance for diverse target groups. 

## 2. Materials and Methods 

This is a qualitative study based on focus group interviews, which are structured group discussions where the moderator allows the participants to influence the content [13]. The study is grounded in constructivist epistemology arguing that scientific knowledge is constructed by the participants, but also taking a stand on the culture of one’s community in an effort to overcome the barriers for participation. Through the use of a number of statements (see Appendix A), and based on DISA (the educational material about slaughter of animals), the moderator steered the discussion so that a number of specific topics were discussed. 

Focus groups as a method facilitates the study of values, motivations, attitudes and behaviours that occur in social interaction and the method is used in situations when meaning is jointly constructed between individual opinions and when a respectful research approach is needed [13].

Two focus groups of 9 and 7 participants met for two hours on 15 January 2016 and 20 January 2016. In total, thirteen individuals were involved in the study and all of the participants from university were conducting research and/or teaching (see Table 1). Two individuals participated in both focus groups and are the authors of this article; another individual was a moderator in both focus groups. Theories were pre-determined and the study was not based on a grounded theory approach. This limited the influence of the participating two authors on the results of the study. However, they were important informants about the creation process.

The meetings were conducted and video recorded through a video conferencing system where it was possible to see and hear every participant. The participants were in four different locations during the first meeting and in three locations during the second meeting. One month before the meetings, the participants received an invitation with some brief information about the planned focus group meetings (see Appendix A). The participants in the first focus group were very familiar with DISA; they were employed by an authority, a university or a slaughterhouse. The majority of the participants in the second focus group were not familiar with DISA; they were employed by the university or by an animal welfare NGO.

The two video recordings, both two hours long, were transcribed in their whole length and these documents included who spoke, what was said, and, in some cases, how it was said. After transcription, the data was anonymized and divided into sections depending on the research questions. All of the excerpts were organized under each research question, across all the respondents, in order to identify consistencies and differences. The analysis was first done by question and later on an individual basis. Thus, participant statements were assigned to the following overarching themes following the research questions: Inclusiveness, Learning and awareness raising, Controversies, and Accuracy and relevance for diverse target groups. These themes were presented under results and discussed in a systemized manner. Citations of relevance were identified as excerpts and citations from the first focus group meeting, which was held in Swedish, were translated into English. Each separate excerpt was treated as a natural unit of analysis and served as a complement to the interpretative thematic analysis. 

## 3. Results and Discussion

The discussions in the focus groups were moderated in order to present arguments for the four approaches to OER: (a) inclusiveness, (b) learning and awareness raising, (c) controversies and (d) accuracy and relevance. The discussions had a tendency to either focus on an argument for open practices or for the instruments to reach openness, and, therefore, each section firstly deals with “Because” and secondly “How” (see Table 2).

Through an interpretive thematic analysis, we were able to identify a rather clear argument for the OER as a boundary object [2,14,15], reflecting a teaching and learning strategy serving a multitude of purposes such as inclusiveness, empowerment and awareness raising. The results also identified challenges with tensions around such a potentially sensitive topic, which difficulties that an OER can help to resolve and which require additional instruments. We will discuss all of these aspects below. The choice to involve the authors in the process was based on the limited number of available experts with deep knowledge of the material, as Sweden is a small country and the possibilities of recruiting experts without any prior involvement in the DISA material was very limited. Still the effect of this involvement may add some doubt to the focus group interpretations, and this choice will hence have to be balanced against the factual information supplied by these persons. 

### 3.1. Inclusiveness

The core of inclusiveness is the social process as a basis for learning and development. One might argue that the only way to manage complex and controversial issues (i.e., contested areas) in contemporary society is through a multi-stakeholder approach featuring collaborative learning. With respect to the inclusive dimension of OER, the participants stressed that: (a) learning is not limited to the dissemination of information, (b) one has to manage the so-called fundamental dilemma between “best available knowledge” and “public participation in knowledge creation”, (c) the knowledge–power dimension must be taken into account when involving stakeholders and, (d) it’s a challenge, but also an opportunity, when involving un-organised groups such as citizens or the public.

Social learning processes are central to the sustainable development of society. The participants discussed the importance of both transparency and empowerment in the development of the OER. In this case, empowerment constitutes more than inclusion, e.g., “the trinity of voices”—to have a say, to be respected and to have real influence. However, social learning for sustainable development is perhaps more than anything else linked to participation and inclusiveness. For example, the DISA material enables people from different backgrounds to get access to the same material, and ideally to give their input to both content and form. Inclusiveness should also be supported from the very beginning and during the development phase. Of course, as a participant stated, there will always be stakeholders with more power (and knowledge) that will affect the outcome (like industry versus NGOs). Altogether this is also a question about the quality of the OER, where a guiding principle should be that we take into consideration many different perspectives on animal welfare issues. This would also result in richer and more socially robust knowledge, something which in itself supports sustainable development.

The participants did not see any way to reach a common understanding between, for instance, the industry and the public, other than to ensure a strong link through public access to the slaughterhouses (as through the videos and photos in DISA). Interestingly enough, the staff at slaughterhouses did not believe openness to be a problem; this is rather a perception among the slaughter industry umbrella organisations. Thus, it is not so much about *if*, but rather *how* one should enable access and *how* to create inclusiveness and common ground. This leads to a discussion on technologies.

Although working with transparency and inclusion as important principles, it was still argued that there is a need for opportunities for learning among closed learning communities (i.e., for slaughterhouse staff). This is not a problem per se, but a reminder that “communities of practice” are an important aspect of social learning and thus sustainable development. Regardless of this, a diversity of technologies to enable people with different preferences to get access and to participate in diverse spaces was stressed as important, as well as ideas on how open educational practices (enabling people to add to the common knowledge base) could increase inclusiveness. This last aspect is part of a bigger discussion in society today on “citizen science”, open data, etc.

Dewey’s [16] claim, almost a century ago, that society is realised through communication gives us incentives to explore dialogue and cultural expression about controversy over the slaughter of animals and the various methods used for this purpose. DISA was created as a “boundary object”, around which different communities communicated and collaborated for different end-uses or end-users. 

This kind of open education is motivated by a belief that students and people in general want to ask questions on their own terms when they learn [17]. A learner’s relational agency, which is the capacity to interact with others in social activities and to respond to complex problems, is driven by a motivation to act related to cultural and/or biological needs, and even more so when emotions and moral concerns are at stake [18].

### 3.2. Learning and Awareness Raising

In general, citizens have very little knowledge about slaughter. The participants argued, in a pluralistic way, in favour of information on slaughter being available to the public because: (a) slaughterhouses are entities that are traditionally closed and slaughter is probably one of the most sheltered activities and, like in all other activities, citizens want transparency, (b) there are a lot of beliefs and naïve views on slaughter and basic knowledge about slaughter of farm animals is a prerequisite to have moral opinions about food, and (c) people with knowledge have an obligation to tell, not least because animals are vulnerable. Thus, there is a case for unbiased and transparent public sharing of information that is available in a way that promotes learning.

Eating meat and hence the necessity of slaughtering animals for the production of meat is a moral concern that is increasingly questioned by a significant number of citizens and consumers [19]. This study highlighted that concerns for the practices of slaughtering animals for food result in tensions between the modern slaughterhouse business operators and other actors in society, something which has been foreseen and partly explained by the change from backyard to industrialised slaughter [20]. It should, however, be acknowledged that the slaughter of animals is not exclusively relevant to meat consumers, as slaughterhouses also contribute to the production of other items, such as hides or pet food, and indirectly are linked also to other aspects such as milk production.

DISA is openly available through a creative commons licence. This means that the material can be downloaded and reused, which one of the participants argued was an important signal value because the industry indirectly demonstrates that they are not off limits—they can withstand close scrutiny. In Sweden, a history of collaboration between academia and slaughterhouses has led to unique access to slaughterhouse facilities, including the opportunity to collect photos and video clips for teaching purposes. One of the participants argued: “That’s not the case in other countries and also the reason why there is an international need for this learning resource”.

Slaughterhouse staff is the main target group for this OER, although it was guided by the principle of promoting universal education. People handling animals at slaughter need to understand the animal’s behavioural needs and natural behaviour in order to recognise why animals must be treated in a specific way. Furthermore, staff should be knowledgeable about the causal link between animal stress and impaired meat quality as further motivation to handle animals with high animal welfare standards. One of the participants stressed that the control body, appointed by the EU Commission to ensure that EU legislation on animal welfare is properly implemented and enforced, has stated that openness indirectly provides DISA with a quality stamp. 

Besides a discussion about diverse target groups and openness, the discussions also focused on the difference between dissemination of information and learning: “Learning requires interactivity, participation and meaningful dialogue”, as one participant put it. This is related to a theory of learning in which interaction, sharing and dialogue are integral parts of the learning process. 

Finally, the focus groups discussed the use of visual material and the importance of the material that needs to be both informative and interesting. In a discussion about the difference of deep versus surface learning, one of the participants commented that a learning resource such as DISA cannot by itself change attitudes and values.

The idea of creating an OER about slaughter contradicts the common practice of actively hiding the knowledge about slaughter practices, as described by Miele [19] (p. 56). The OER provides access for everyone to the available information about slaughter. There is consensus in the scientific community that animals are sentient beings with the ability to suffer and that society has a duty to end cruel farming systems and trades and practices that inflict unnecessary suffering on animals [21]. This can be interpreted as one of our critical duties and thus part of education for a sustainable development.

Education for sustainable development is, according to UNESCO [22], when “integrating key sustainable development issues into teaching and learning…and utilising new forms of learning that require multi-stakeholder interaction, meaning making, negotiation, dealing with competing claims, handling diversity of perspectives (cultural, disciplinary, socio-economic, etc.) and the resolving of real issues as they emerge in everyday life at home, in the university itself, in the community or in the work-place”. It should be stressed, however, that the aim of an OER is rarely to achieve consensus, but hopefully a common understanding or at least a useful platform for handling inevitable diversity of perspectives in a controversial topic such a slaughter.

### 3.3. Controversies

It is generally accepted that controversy and tension can be a catalyst for change. The participants in the focus groups discussed that tension related to open knowledge about slaughter could: (a) close the knowledge gap between providers and consumers of knowledge, (b) blur the difference between the roles of providers and consumers of knowledge, (c) both resolve and create conflicts between cultures.

Members of the slaughter umbrella organisations had a fear that openly available video clips and images from the slaughterhouses could possibly create criticism of the slaughter of animals. This fear created a conflict between the umbrella organizations, the creators and the funder. However, the launch did not result in any negative reactions, and, so far, DISA has not created any conflicts once having been launched. Instead, DISA can be used to deal with conflicts, as one of the focus group participants demonstrated by using the example of a single, open and transparent source being referenced instead of having to rely on several different sources. 

Potential pitfalls in controversial issues can be avoided by sticking to the facts. The focus group participants argued that it was never wrong to educate people and that DISA is a way of closing the gap between industry and consumers/citizens—giving people the tools to start speaking the same language. The current version of DISA was regarded as non-biased among the participants in the focus groups. One participant representing the slaughterhouses argued that “DISA is neutral and depicts slaughter as it is. It lifts how to handle animals as the best way to promote good animal welfare at slaughter without finding scapegoats”. 

However, an open resource can create conflicts in spite of being neutral, since slaughter is an emotional topic and often emotion in itself hampers a constructive discussion, one of the participants pointed out. “It is not a question about if the public can understand the information but rather if it is too emotional and if it will upset the learner”. Another focus group participant had a different view of how an open resource could handle conflicts: “DISA is valuable in terms of bringing society closer to food production and animal welfare practices and so on… We as scientists and creators are representing the interest of the animals”. 

The involvement of many people with diverse backgrounds in both the creation and review of DISA was deliberate in order to increase inclusiveness and thus control quality, as exemplified by the inclusion of representatives from religious groups that scrutinized the sections on kosher and halal slaughter. However, one of the participants argued that opening up a discussion with the public on religious slaughter may generate a lot of work because of the risk that someone may want to sabotage the material.

Furthermore, it is well known that the way science is presented can also lead to conflicts. One participant argued that a learning resource might generate tension if it has authorities as the author and content showing a current practice of handling animals at slaughter, which is still within the legal limits according to the animal welfare legislation, but at the same time inevitably negatively affects animal welfare to a certain extent. “What a lot of people see online—they don’t think what they see is legal. When they get to know that this is legal and that there is science behind it, it can create tension”.

The focus groups concluded that an OER may be perceived as ‘neutral’ and purely informative by some, but, by others, it may be perceived as quite sensitive, offensive and upsetting. For a topic such as slaughter, there are clear cultural and religious differences with respect to what animal species (if any) are to be slaughtered for food, how animals are slaughtered (with or without stunning, for example) and how this is to be done in the best possible way, from different perspectives. Hence, when working with an OER on these topics, it is important to acknowledge the risk of intentional sabotage to the material such as campaigns by religious groups, by activists and extremists and, not least, tension between these stakeholders. Given these risks, which were carefully analysed prior to the creation and launching of the DISA material, the negative interactions were limited. On the contrary, it can be hypothesized that the openness of the material can actually help to resolve tension between different stakeholders.

The results also included a discussion of the benefits and concerns related to the use of video and photos in the OER. In the literature, it has been shown that multimodality makes the experience richer and that each learner can get the same view as all the other learners and can view the process many times until they have mastered it (i.e., self-directed learning) [23]. In the case of slaughter, the processes are very complex and both the behaviour of the abattoir staff and the animal and how these behaviours are interconnected needs to be understood. This is very difficult to comprehend when visiting a slaughter house. It is also difficult for groups of students to observe animals; therefore, the use of photos and videos is critical.

### 3.4. Accuracy and Relevance 

The power of OER lies in the five Rs, which are the rights to retain, reuse, revise, remix and redistribute the OER with an open licence [24]. Generally speaking, it is impossible to revise, remix, or redistribute an openly licensed work unless one possesses a copy of the work. Thus, the right to retain, read or control an original version of the content underlies open licenses such as creative commons, the licence attached to DISA. 

Both focus groups discussed the need to update the material as a result of (a) new scientific discoveries, (b) new practices and (c) normative changes in society.

As expressed by one of the participants representing a university: “It is never a final product…it can always be amended depending on the reaction we get from the users being slaughterhouses, students or citizens”. Another participant argued: “It is the authorities and the government that finally decide what is an acceptable animal welfare level, but they do that by asking the stakeholders…I mean the industry, the animal welfare organisations, the consumer organisation… in the end it is actually the ones that writes the regulation that decide but their task is to kind of feel what is the society’s moral standing…what is the ethical levels of the society”.

This discussion included suggestions on how to revise DISA. Today, users can suggest improvements and point at outright errors by e-mail. One suggestion was to introduce moderated discussion forums for different target groups such as slaughter staff or the general public. A majority of the participants believed that a Wikipedia-based OER in such a controversial issue would cause problems, since it would require careful monitoring and that DISA is not expected to get enough users to ensure that any incorrect or offensive modifications, additions or alterations made by others are rapidly removed.

When creating OER for highly relevant societal issues, and especially when dealing with ethically sensitive issues, the credibility and relevance of the content is particularly important. The moderator for the focus groups summarised the discussion with: “DISA is a material that the public can relate to and, at the same time, it has so much depth that it makes sense for the primary target group (the slaughterhouses)”.

DISA was originally drafted for the Swedish context; however, some of the visual material originates from other countries. The English version of DISA, which is partly launched, has an EU perspective because it is based on the EU legislation. One of the participants argued “Already trying to create a generally useful teaching material related to slaughter in a Swedish context will be a handful; trying to use this in other cultures it might create problems”.

One of the participants of the focus groups questioned if an OER in itself could change the participants’ attitudes and values, and the authors agree that well-established values are not easily changed by one source only. However, recent research has shown that users’ attitudes can be changed through the use of educational resources on the Internet. In an EU-funded study, users of a digital learning resource about farm animal welfare changed their attitudes to the handling of the farm animals as they were gaining greater knowledge [25]. Further research should focus on impact studies and analysis of attitudes and consumer behaviour before and after DISA being used by specifically selected target groups in different countries. Animal welfare resilience has recently been introduced as the ability to tolerate (critique of what the society has agreed to be good animal welfare) and even benefit from distress and to sustain without major changes [26], and the DISA material can act as an example of this phenomenon.

Further development of DISA with a shared medium that enhances the conversational framework to also enable learners to solve problems would be an improvement for democracy and sustainability. It is a way to respond to the greater ethical obligation related to educational materials that are electronically copied and transferred around the world [27]. However, publishing an OER on slaughter will inherently lead to conflict [28] and such a development is expected to have an impact on the incidence of conflicts related to the topic in question, and increase the need to moderate the conversation. When subjects are focused, OERs about these narrow subjects cannot be expected to be self-repairing as one of the participants in the focus groups mentioned. Hence, not limiting these possibilities may have major drawbacks, such as the risks of introduction of false information, fake data and offensive content into the text and Appendix A. These two aspects have to be carefully weighed, and the outcome will at least partly depend on the resources available.

## 4. Conclusions

This case study of controversy and cross-cultural perspectives is an illustrative example targeting learning and awareness raising for citizens and indirectly animal welfare resilience. Based on this limited study, we suggest that an open educational resource, in addition to contributing to learning and awareness raising, can create dialogue about concerns for animal welfare through built-in inclusiveness.

Furthermore, we have identified an inherent value of providing this knowledge in an open environment, and, currently, no major negative effects have been reported. Nevertheless, we have identified some risks, should the internet-based learning resource be open for interactive additions without oversight, as the topic is emotionally and politically sensitive, not least in relation to religious slaughter.

## Figures and Tables

**Table 1 animals-07-00101-t001:** The thirteen participants in the two focus groups.

Focus Group	Gender (M/F)	Employment	Co-Creators of DISA	Users of DISA	Co-Authors of Article
1	F	Authority	No	Yes	No
1	F	University *	Yes	Yes	No
1	M	Slaughterhouse	No	Yes	No
1	F	Slaughterhouse	No	Yes	No
1	F	Authority	Yes	Yes	No
1	M	University	Yes	No	No
1 + 2	M	University	No	No	No
1 + 2	F	University *	Yes	Yes	Yes
1 + 2	F	University *	Yes	Yes	Yes
2	F	University *	No	No	No
2	F	NGO	No	No	No
2	M	University	No	No	No
2	M	University	No	No	No

DISA—an acronym for “Djurvälfärd i samband med Slakt och annan Avlivning” in English: “Animal welfare at slaughter and killing”. * These individuals were also lecturers on animal welfare.

**Table 2 animals-07-00101-t002:** An illustration of how the coding was done.

.	Quote as an Example of “Because”	Quote as an Example of “How”
Inclusiveness	“It is important to have both citizens and scientists to inform each other what is fair and what is cruel”	“The slaughterhouses and the umbrella organizations, they were very active; they shared photo material and corrected text about practical handling details”
Learning and awareness raising	“It is never wrong to educate people… there is a gap between the industry and the consumers and this [DISA] is actually a way of trying to close that gap and start to talk the same language”	“I guess that learning that leads to change is far more difficult…unless it’s an active process…”
Controversies	“In opening up the slaughter houses on the Internet we are also opening up science…because science is informing the practice in the slaughterhouses…if this is what science says about best practice and look it’s still very gruesome it could lead to some kind of distrust and questioning of science behind it”	“Discussions in open fora very rapidly become polarized and not so civilised. There is a risk of conflicts which needs moderating”
Accuracy and relevance	“During creation and management, we have to include people since we are not the only ones to decide what is important. It is never a finalised product…it can always be amended, changed, depending on the reaction we get from the users being slaughter house, students or citizens”	“Should DISA have both active and passive quality assurance? With active update, I mean after five years of operation return to the groups that were initially active in the creation...the religious groups, the researchers and the industry”

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
