# Peer review of "Open Knowledge about Slaughter on the Internet—A Case Study on Controversies"

_animals, 2017, doi:10.3390/ani7120101_

Round 1

Reviewer 1 Report

This paper explores industry and academic stakeholder perceptions of an open educational resource on animal slaughter. These types of OER are becoming a common aspect of animal welfare research, and I believe this paper should be of great interest to the Animals readership. 

The authors have attempted to make their qualitative methods clear to this audience, who are predominantly more familiar with quantitative research, and this should be commended. However they have missed out a few of the more pertinent details, for example, the authors appear to have participated in the focus groups as participants, and this needs to be openly discussed so readers can judge whether the authors may have had an undue influence on the topics. (Typically I would not write up a paper I had been a participant in)

The written English needs to be improved and I've highlighted a few areas that were not clear to me, however the revision would greatly benefit by sending it to a native English speaker for proof reading. 

I would suggest that the authors combine their results and discussion so that the emergent themes can be discussed in light of the literature. I believe the authors are coming a constructivist viewpoint and so some of the results, e.g. the conflict between how 'sensitive' academics and the public perceived the resource to be, could benefit from a more theoretical discussion of why this matters in education research.

Also - I don't think the title shows off the work to its best advantage. Either be more explicit about the controversies in the text, or change the title. 

Line 12: Suggest change to this long sentence: Animal products are consumed by a large majority of the global population, and yet public knowledge about animal handling and welfare during the slaughter process is limited.

Line 15: suggest change: An open educational resource about animal slaughter has been openly available on the internet since 2012. Titled “animal welfare..”

Line 20: describe how analysed (and common themes were explored?)

Lines 22-24: Sentence is too long.  Increase not increased

Lines 25: Suggest change: Information about animal slaughter is often perceived as inflammatory and therefore not made widely available.

Line 25: Can you demonstrate how it has created and resolved tension? This is a result that is not backed up with a finding in the abstract.

Line 32: no comma

Line33: attitudes towards animal slaughter?

Line 33-35: This is not clear to me (an education researcher) and so won’t be clear to readers of Animals. Rephrase without the educational terminology

Introduction General Comment:

Much of the introduction is spent talking about the development of the course. It’s always tricky in educational papers to decide where that should go. I would move the discussion of boundary objects to the start of the introduction, and then talk about the course development towards the end of the introduction so you are starting with the theoretical underpinnings and then exploring the practical aspects.

Line 51: “animal slaughter is a contested area” is an odd phrase – controversial might be the word you’re looking for

Line 59: Learning objectives are a key part of instructional design and assessment of teaching (see Schoenfeld-Tacher & Sims 2013, Journal of Veterinary Medical Education). Learning objectives should be first on the list, they are more important than the number of webpages. It may also be worth highlighting to readers of Animals why learning objectives are so important

Line 61: Do you mean ‘agents’ instead of ‘actors’?

Line 62: is controversial because such pictures..

Line 63: The slaughter industry’s umbrella organisation were concerned that images from the course would be used by animal rights groups to discredit abbatoirs and the industry in general. This concern was not voiced by the individual slaughter houses.

Line 66-68: Completely unclear

General M&M Comment:

You take a lot of time describing the practicalities of focus groups, which I think is good for this journal as the readers may not have run one before. However, I do think you miss some important information (e.g. did you pre-form your theories, did you have a pre-identified epistemiological stance, if the theories were data-generated was this a grounded theory style approach?)

Line 104: Ethical permission for study?

Line 114: Was one of the authors a focus group participant? E.g., contributed to the data generated outside of a moderator role?  I am a little uncomfortable with this and would like some explanation of why it was done and possible impacts on the data to allow readers to judge whether this was appropriate

Line 125: If it would not identify your participants, I would be interested to know what kinds of roles they served at the university, e.g. were they animal welfare lecturers?

Lines 127-131: This way of separating the focus group data makes it harder to look at concept linkages – just a thought for future studies.

Lines 132-138: The formation and codifying of themes is a really important topic in qualitative research and I find is actually quite innately familiar to ethologists – you could present this information in a sort of ethogram like table with a definition of the theme and an example of it in action (a quote) and it will be very clear to the readers of Animals

Line 144: Not clear what you mean by a time factor in your analysis?

Line 156: This is a very constructivist viewpoint (most OER stuff is constructivist I find). I really think you need to identify your epistemology and ontology in the methods

Line 256: Need a citation for the 5 Rs (especially animal welfare is more familiar with the 3rs)

Line 265: This sounds like a university worker to me – I tend to find industry prefers final products. Throughout you need to be more clear the origin of your claims (I’d also like quotes but I suspect they may be in Swedish which I am not fluent in!)

Lines 307-325: I think it is a bit misleading to frame this as a problem for consumers of meat – slaughterhouses provide us far more resources than meat, and I would really like this discussion to be widened to demonstrate that this is not an optional part of a commercial society.

Line 312: You need a good reference for this ‘hiding slaughter’ idea.

Line 329-348: This perceived sensitivity of the resource is a really good point – and something I’ve encountered myself in my own research. I would place more emphasis on this point (you could even discuss how constructed knowledge for academics does not reflect how that knowledge is perceived by the public!) This is another good argument for combining the results and discussion I think as I’d like this discussion alongside the results it talks about

Author Response

We are grateful for excellent comments from this reviewer. Please see attachment.

Reviewer 2 Report

I found this an interesting study. I think it could benefit from some English language editing and that this should take care with the introduction of the more specialist terminology such as 'boundary object' etc. I don't know of a similar study elsewhere, but it is topical in connection with calls for video monitoring of slaughterhouses in order to ensure standards.

Author Response

Thank you for the comments on this manuscript.      

The manuscript has now been scrutinized by a native English speaker.

Reviewer 3 Report

Originality / Novelty

The research question addressed in this paper is original, as I understand it, looking to explore the tensions created between industry and science in increasing increased transparency surrounding how food is produced. This is a pertinent question as high quality and objective resources such as DISA are created with a great deal of resource but may not able to be disseminated openly or supported by industry. It is important to understand the types of barriers those who work to produce high quality objective animal welfare resources face in developing and disseminating their products.

The results as reported provide some advances in knowledge in relation to the research question. However, this is limited at present to some extent by some methodological concerns regarding the perhaps limited range of participants, as well as some lack of clarity in English language used in the paper.

Significance

The paper does not include a very clear or explicit hypothesis regarding the status of the DISA resource so it could be made clearer. At this stage, it is not clear if the results can be considered as having been interpreted appropriately. I do not see how the study enables the conclusion that “an open educational resource in addition to contributing to learning and awareness raising can also induce dialogue about animal welfare and contribute to animal welfare resilience”. There are four main reason for this:

1)    Any educational resource that is pitched at Slaughter house workers (who need a very technical knowledge and most likely a pragmatic/kinesthetic learning styles?)  is unlikely to also be appropriately pitched at the suitable level for consumers, who will not have familiarity with technical terms for example and may not have the same learning styles. Additionally, how can what is being learned be gaged without some form of assessment? Does the DISA tool include summative or formative assessments at all (I would expect it to for Slaughter workers) and what inclination would there be for consumers to complete assessments if it did?

2)    Some restrictions in information about the participants involved in the focus groups. For example, to conclude that the DISA resource will be effective in making citizens more knowledgeable requires all its target audiences to be involved in the focus groups. Were citizens included in the focus groups though?  This is not clear from Table 1 which could be improved by including information about if the participant works in research and / or teaching related to an animal science or another area of expertise or purely administrative staff. Additionally, because this resource concerns slaughter, it would be best to clarify if possible the religious beliefs of the participants so the reader can gage if the relevant stakeholders in religious terms have been included in the research. I would also argue that some members of the general public / consumers (a key proportion of society along with NGOs, Authorities, etc.) should be included in focus groups as opposed to just university workers. This would be required to make more generalized statements about the DISA resource being on the boundary between academia and ‘society’.

3)    There are some possible assumptions surrounding consumer interest in learning more about slaughter. The phenomenon of ‘cognitive dissonance’ is relevant here. Consumers on the one hand are concerned for animal welfare, as reported in the Eurobarometer, but this does not predict being prepared to stop consuming meat or the act of actively seeking to learn more and expose oneself to the technicalities of slaughter. There is a growing body of evidence that show that even learning about and increasing knowledge about  animal welfare does not predict a change in behavior.  Additionally, consumers may be happier putting their faith in a labelling system that confirms that an animal product has been produced with good standards animal welfare rather than delving into the detail of the slaughter process themselves.

4)    Some limitations in English language used make it difficult to understand sentences

In this respect, for now it would seem that the authors are perhaps overly confident in the conclusions from this study.  I would consider this research more of a pilot study that would warrant further research with wider audiences.

Quality of presentation 

The Introduction to the paper lacks some flow between paragraphs about starting at line 83 and 89. If the concept of boundaries can be introduced in a more user friendly format the paper would benefit. It would open up understanding more for audiences that do not have familiarity with social science concepts and research approaches. Perhaps including some subheadings that encapsulate the issue addressed in each paragraph would help. Additionally, the terms ‘inclusiveness’, ‘learning’, ‘awareness raising’, ‘controversies’, ‘accuracies’ and ‘relevance for diverse target groups’ are introduced in the Introduction but is clarity was provided as to what these terms mean conceptually in the context of the paper this would help. Further breakdown of what phrases and quotes the participants offered would help to highlight the authors analyses and the readers subsequent understanding.

Scientific soundness[NC4] 

For the audience that are not familiar with qualitative or social sciences research the materials and methodology is short of explanations and justification.  The way that the data and analyses are presented are limited in that for any of the audience that do not have a grounding in social sciences there is very little structure. Including a figure that outlines the information provided to participants on recruitment about the focus groups (line 122) and the questions used in the focus groups. The paper would benefit from the inclusion of some scientific rationale for the use of qualitative approaches and forms of analysis (interpretive thematic analysis) and the systematic steps taken as part of the analysis. As in its current format it would not be easy for another research to reproduce carrying out the study.

Interest to readers

The research is interdisciplinary in terms of its focus on the trajectory of animal welfare science (slaughter), education (the DISA resource) and social science (focus on boundary objects and the qualitative research approach). It will therefore appeal to a diverse audience within academia. It will however also appeal to wider audiences according into the stakeholders included in the study (e.g. NGOs, the good production / slaughter industry, authorities).  The cross-cultural perspectives on the issue explored are acknowledged but the relevance of this research in this respect could be explored in more depth (also cross related to comments about the range of participants included in focus groups).

Overall merit

With some further work on the paper there is most definitely an overall benefit to publishing this work as publication of research on this issue can help to advance current knowledge on how to move dialogue forwards between academia and research, and between food production and consumer awareness. However, it is not clear from the paper in its current format how rigorous the research methods used and analyses performed in this instance are.

English level 

Approximately 75% of the papers is written in appropriate English language. There are several instances of plural and singular forms of verbs are misused in places (e.g. ‘is’ should be used instead of ‘are’ (line 12). Some verbs are used ambiguously (e.g. line 141 ‘as a means to “handle”; line 277 ‘Self-cleaning”– these verb does not fit with the sentence and it is not clear what is meant. Perhaps instead ‘facilitate’ or ‘stimulate’). There are various instances of sentences that don’t have a clear meaning (e.g. line 365 ‘However….moderating”  line 367). Line 374; This becomes particularly complicated where attempts have been made to explain certain terms (e.g. ‘Welfare resilience’) and concepts (e.g.lines 148 - 154) results and conclusions.

Author Response

(The authors gave the same response as above.)
